# ShapeFlow:
# Learnable Deformations Among 3D Shapes

**Chiyu "Max" Jiang**[*]
UC Berkeley
chiyu.jiang@berkeley.edu

**Jingwei Huang**[*]
Stanford University
jingweih@stanford.edu

**Andrea Tagliasacchi**
Google Brain
taglia@google.com

**Leonidas Guibas**
Stanford University
guibas@stanford.edu

## Abstract

We present `ShapeFlow`, a flow-based model for learning a deformation space for entire classes of 3D shapes with large intra-class variations. `ShapeFlow` allows learning a multi-template deformation space that is agnostic to shape topology, yet preserves fine geometric details. Different from a generative space where a latent vector is directly decoded into a shape, a deformation space decodes a vector into a continuous flow that can advect a source shape towards a target. Such a space naturally allows the disentanglement of geometric style (coming from the source) and structural pose (conforming to the target). We parametrize the deformation between geometries as a learned continuous flow field via a neural network and show that such deformations can be guaranteed to have desirable properties, such as bijectivity, freedom from self-intersections, or volume preservation. We illustrate the effectiveness of this learned deformation space for various downstream applications, including shape generation via deformation, geometric style transfer, unsupervised learning of a consistent parameterization for entire classes of shapes, and shape interpolation.

## 1 Introduction

Learning a shared representation space for geometries is a central task in 3D Computer Vision and in Geometric Modeling as it enables a series of important downstream applications, such as retrieval, reconstruction, and editing. For instance, *morphable models* [1] is a commonly used representation for entire classes of shapes with small intra-class variations (i.e., faces), allowing high quality geometry generation. However, morphable models generally assume a *shared* topology and even the same mesh connectivity for all represented shapes, and are thus less extensible to general shape categories with large intra-class variations. Therefore, such approaches have limited applications beyond collections with a shared structure such as humans [1, 2] or animals [3].

In contrast, when trained on large shape collections (e.g., ShapeNet [4]), 3D generative models are not only able to learn a shared latent space for entire classes of shapes (e.g., chairs, tables, airplanes), but also capture large geometric variations between classes. A main area of focus in this field has been developing novel geometry decoders for these latent representations. These generative spaces allow the mapping from a latent code $z \in \mathbb{R}^c$ to some geometric representation of a shape, examples being voxels [5, 6], meshes [7, 8], convexes [9, 10], or implicit functions [11, 12]. Such latent spaces are

---

[*]Equal Contribution.

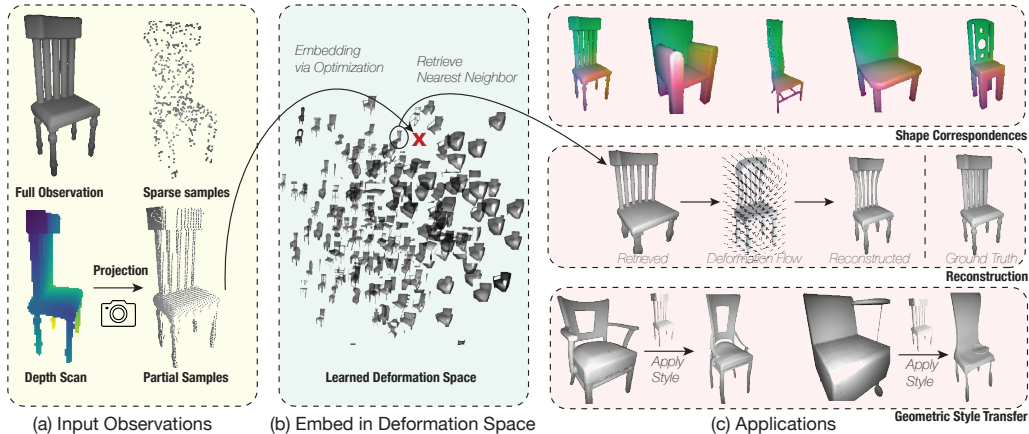

Figure 1: Schematic for learning a deformation space using `ShapeFlow`. (a)Our input is either a sparse point cloud, or a depth map converted into a point cloud. (b) The visualization of the learned latent embedding (2D PCA) of sample shapes in the training set. `ShapeFlow` learns a geometrically meaningful embedding of geometries based on deformation distances in an unsupervised manner. (c) The unsupervised deformation space facilitates various downstream applications, including shape correspondences, reconstruction, and style transfer.

generally smooth and allow interpolation or deformation between arbitrary objects represented in this encoding. However, the shape generation quality is highly dependent on the decoder performance and generally imperfect. While some decoder architectures are able to produce higher quality geometries, auto-encoded shapes *never exactly match* their inputs, leading to a loss of fine geometric details.

In this paper we introduce a different approach to shape generation based on continuous flows between shapes that we term `ShapeFlow`. The approach views the shape generation process from a new perspective – rather than learning a generative space where a learned decoder $\mathcal{F}_\theta$ directly maps a latent code $z_i \in \mathbb{R}^c$ to the shape $X_i$ as $X_i = \mathcal{F}_\theta(z_i)$, `ShapeFlow` learns a *deformation space* facilitated by a learned deformer $\mathcal{D}_\theta$, where a novel shape $X_{i \leftarrow j}$ is acquired by deforming one of many possible *template shapes* $X_j \in \mathcal{X}$ via this learned deformer: $X_{i \leftarrow j} = \mathcal{D}_\theta(X_j; z_j, z_i)$, where $z_i, z_j \in \mathbb{R}^c$ are the latent codes corresponding of $X_i$ and $X_j$.

This deformation-centric view of shape generation has various unique properties. First, a deformation space, compared to a generative space, naturally disentangles geometry *style* from *structure*. Style comes from the choice of source shape $X_j$, which also includes the shape topology and mesh connectivity. Structure includes the general placement of different parts, such as limb positioning in a human figure (i.e., pose), height and width of chair parts etc. Second, unlike template-based mesh generation frameworks such as [13, 14, 7], whose generated shapes are inherently limited by the the template topology, a deformation space allows a multi-template scenario where *each* of the source shapes $X_j \in \mathcal{X}$ can be viewed as a template. Also, unlike volumetric decoders that require a potentially computationally intensive step for (e.g., querying a large number of sample points), `ShapeFlow` directly outputs a mesh (or a point cloud) through deforming the source shape. Finally, by routing the deformations through a common waypoint in this space, we can learn a *shared template* for all geometries of the same class, despite differences in meshing or topology, allowing unsupervised learning of dense correspondences between all shapes within the same class.

The learned deformation function $\Phi_\theta^{ij}$ deforms the *template* shape $X_j$ into $X_{i \leftarrow j}$ so that it is geometrically close to the target shape $X_i$. Our deformation function is based on neurally parametrized 3D vector fields or *flows* that locally advect a template shape towards its destination. This novel way of modeling deformations has various innate advantages compared to existing methods. We show that deformation induced by a flow naturally prevents *self-intersections*. Furthermore, we demonstrate that we can parametrize a divergence-free flow field effectively using a neural network, which ensures *volume conservation* during the deformation process. Finally, `ShapeFlow` ensures path invertibility ($\Phi_\theta^{ij} = (\Phi_\theta^{ji})^{-1}$), and therefore also identity preservation ($X_i = \Phi_\theta^{ii}(X_i)$). Compared to traditional deformation parameterizations in computer graphics such as control handles [15, 16]

and control cages [17, 18, 19], `ShapeFlow` is a flow-model realized by a neural network, allowing a more fine grained deformation without requiring user intervention.

In summary, our main contributions are:

1. We propose a flow-based deformation model via a neural network that allows exact preservation of identity, good preservation of local geometric features, and disentangles geometry style and structure.

2. We show that our deformations by design prevent self-intersections and can preserve volume.

3. We demonstrate that we can learn a common *template* for a class of shapes through which we can derive dense correspondences.

4. We apply our method to interpolate shapes in different poses, producing smooth interpolation between key frames that can be used for animation and content creation.

## 2   Related work

Traditionally, shape representation in 3D computer vision roughly falls into two categories, *template-based* representation and *template-free* representation. In contrast, `ShapeFlow` fills a gap in between – it can be viewed as a *multi-template space*, where the source topology can be based on any of the training shapes, and where a very general deformation model is adopted.

**Template-based representations.**  These methods generally assume a fixed topology for all modelled geometries. Morphable models [1] is a commonly used representation for entire classes of shapes with very small intra-class variations, such as faces [1, 20, 21, 22, 23, 24], heads [25, 26], human bodies [27, 28, 2], and even animals [3, 29]. Morphable models generally assume a shared topology and even the same mesh connectivity for all represented shapes, which restricts its use to few shape categories. Recently, neural networks have been employed to generate 3D shapes via morphable models [30, 31, 14, 32, 33, 3, 29]. Some recent work has extended the template-based approach to shapes with larger variations [13, 7, 34, 35], but the generated results are polygonal meshes that often contain self-intersections and are not-watertight.

**Template-free representations.**  These methods generally produce a volumetric implicit representation for the geometries rather than directly representing the surface under certain surface parameterizations, thus allowing the same model to model geometries across different topologies, with potentially large geometric variations. Earlier works in this line utilize voxel representations [36, 37]. Recently, the use of continuous implicit function decoders [38, 39, 11] has been popularized due to its strong representation capacity for more detailed geometry. Similar ideas are extended to represent color, light field, and other scene related properties [40, 41], and coupled with spatial [42, 43] or spatio-temporal [44] latent grid structures to extend to larger scenes and domains. Still, these approaches lack the fine structures of real geometric models.

**Shape deformation.**  Parametrizing the space of admissible deformations in a set of shapes with diverse topologies is a challenging problem. Directly predicting offsets for each mesh vertex with insufficient regularization will lead to non-physical deformations such as self-intersections. In computer graphics, geometry deformation is usually parameterized using a set of deformation handles [15] or deformation cages [17, 18, 19]. Surface-based energies are usually optimized in the deformation process [45, 46, 16, 47] to maintain rigidity, isometry, or other desired geometric properties. Similar to our work, earlier work by [48] proposed deforming objects using time-dependent divergence-free vector fields, though the deformations are not learnable. More recently, learned deformation models have been proposed, directly predicting vertex offsets [49], control point offsets [50], or control cage deformations [51]. Different from our end-to-end deformation setting, the graphics approaches are typically aimed at interactive and incremental shape editing applications.

**Flow models.**  Flow models have traditionally been used in machine learning for learning generative models for a given data distribution. Some examples of flow models include RealNVP [52] and Masked Auto-Regressive Flows [53]; these generally involve a discrete number of learned transformations. Continuous normalizing flow models have also been recently proposed [54, 55], and our method is mainly inspired by these works. They create bijective mappings via a learned advection process, and are trained using a differential Ordinary Differential Equation (ODE) solver. PointFlow [56] and OccFlow [57] are similar to our approach in using such learned flow dynamics for modeling geometry. However, PointFlow [56] maps point clouds corresponding to geometries

to a learned prior distribution while `ShapeFlow` directly learns the deformations function between geometries, bypassing a prior distribution and better preserves geometric details. OccFlow [57] only models the temporal deformation sequence for one object, while `ShapeFlow` learns a deformation space for entire classes of geometries.

# 3 Method

Consider a set of $N$ shapes $\mathcal{X} = \{X_1, X_2, \cdots, X_N\}$. Each shape is represented by a polygonal mesh $X_i = \{\mathcal{V}_i, \mathcal{E}_i\}$, where $\mathcal{V}_i = \{v_1, v_2, \cdots, v_{n_i}\}$ is an ordered set of $n_i$ points that represent the vertices of the polygonal mesh. For each point $v \in \mathcal{V}_i$, we have $v \in \mathbb{R}^d$. $\mathcal{E} = \{e_1, e_2, \cdots, e_{m_i}\}$ is a set of $m_i$ polygonal elements, where each element $e \in \mathcal{E}_i$ indexes into a set of vertices $v \in \mathcal{V}_i$. For one-way deformations, we seek a *mapping* $\Phi_\theta^{ij} : \mathbb{R}^d \mapsto \mathbb{R}^d$ that minimizes the geometric distance between the deformed source shape $\Phi_\theta^{ij}(X_i)$ and the target shape $X_j$:

$$\underset{\theta}{\arg\min} \quad \mathcal{C}(\Phi_\theta^{ij}(X_i), X_j), \tag{1}$$

where $\mathcal{C}(X_i, X_j)$ is the *symmetric* Chamfer distance between two shapes $X_i, X_j$. Note the mapping operates on the vertices $\mathcal{V}_i$, while retaining the mesh connectivity expressed by $\mathcal{E}_i$. As in previous work [58, 38], since mesh-to-mesh Chamfer distance computation is expensive, we proxy it using the point set to point set Chamfer distance between uniform point samples on the meshes. Furthermore, in order to learn a symmetric deformation space, we optimize for maps that minimize the *symmetric* deformation distance:

$$\underset{\theta}{\min} \quad \mathcal{C}(\Phi_\theta^{ij}(X_i), X_j) + \mathcal{C}(X_i, \Phi_\theta^{ji}(X_j)). \tag{2}$$

We define such maps as an advection process via a *flow* function $\boldsymbol{f}_\theta(x(t), t)$, where we associate intermediate deformations with an interpolation parameter $t \in [0, 1]$. For any pair of shapes $i, j$:

$$\Phi_\theta^{ij}(\boldsymbol{x}_i \in X_i) = \boldsymbol{x}_i(1), \quad \boldsymbol{x}_i(T) = \boldsymbol{x}_i + \int_0^T \boldsymbol{f}_\theta^{ij}(\boldsymbol{x}_i(t), t)\, dt. \tag{3}$$

**Intersection-free deformations.** Not introducing self-intersections is a key property in shape deformation, since self-intersecting deformations are not physically plausible. In Proposition 1 (**supplementary material**), we prove that this property is *algebraically* satisfied in our formulation. Note that this property holds under the assumption of perfect integration. Errors in numerical integration will lead to its violation. However, we will empirically show in Sec. C.2 (**supplementary material**) that this can be controlled by bounding the numerical integration error.

**Invertible deformations.** For any pair of shapes, it would be ideal if performing a deformation of $X_i$ into $X_j$, and then back to $X_i$, would recover $X_i$ exactly. We want the deformation to be *lossless* for identity transformations, or, more formally, $\Phi^{ij}(\Phi^{ji}(\mathbf{x})) = \mathbf{x}$. In Proposition 3 (**supplementary material**), we derive a condition on $\boldsymbol{f}_\theta$ that is *sufficient* to ensure bijectivity $\forall t \in (0, 1]$:

$$\boldsymbol{f}_\theta^{ji}(\boldsymbol{x}, t) = -\boldsymbol{f}_\theta^{ij}(\boldsymbol{x}, 1 - t), \quad t \in (0, 1]. \tag{4}$$

## 3.1 Deformation flow field

At the core of the learned deformations (3) is a learnable flow field $\boldsymbol{f}(\boldsymbol{x}, t)$. We start by assigning latent codes $\boldsymbol{z}_i, \boldsymbol{z}_j \in \mathbb{R}^c$ to the shapes $i, j$, and then define the flow as:

$$\boldsymbol{f}_\theta^{ij}(\boldsymbol{x}, t) = \underbrace{\boldsymbol{h}_\eta(\boldsymbol{x}, \boldsymbol{z}_i + t(\boldsymbol{z}_j - \boldsymbol{z}_i))}_{\text{flow function}} \cdot \underbrace{\boldsymbol{s}_\sigma((\boldsymbol{z}_j - \boldsymbol{z}_i)/||\boldsymbol{z}_j - \boldsymbol{z}_i||_2)}_{\text{sign function}} \cdot \underbrace{||\boldsymbol{z}_j - \boldsymbol{z}_i||_2}_{\text{flow magnitude}}, \tag{5}$$

where $\theta = \{\eta, \sigma\}$ are trainable parameters of a *neural network*. Note the same deformation function can be *shared* for all pairs of shapes $(i, j)$, and that this flow satisfies the invertibility condition (4).

**Flow function.** The function $\boldsymbol{h}_\eta(\boldsymbol{x}, \boldsymbol{z})$, receives in input the spatial coordinates $\boldsymbol{x}$ and a latent code $\boldsymbol{z}$. When deforming from shape $i$ to shape $j$, the latent code $\boldsymbol{z}$ linearly interpolates between the two endpoints. $\boldsymbol{h}_\eta(\cdot) : \mathbb{R}^{d+c} \mapsto \mathbb{R}^d$ is a fully-connected neural network with weights $\eta$.

**Sign function.** The sign function $s_\sigma(z)$, receives the normalized direction for the vector from $z_i$ to $z_j$. The sign function has the additional requirement that it be symmetric, which can be satisfied either by *fully-connected neural networks* with learnable parameters $\sigma$, with zero bias and symmetric activation function (e.g., `tanh`), or by construction via the hub-and-spokes model of Section 3.2.

**Flow magnitude.** With this regularization, we ensure that the distance within the latent space is directly proportional to the amount of required deformation between two shapes, and obtain several properties:

- *Consistency* of the latent space, which ensures deforming half way from $i$ to $j$ is equivalent to deforming all-way from $i$ to the latent code half-way between $i$ and $j$:

$$\int_0^\alpha \boldsymbol{f}_\theta^{ij}(\boldsymbol{x}(t), t)dt = \int_0^1 \boldsymbol{f}_\theta^{ik}(\boldsymbol{x}(t), t)dt, \text{ where } \boldsymbol{z}_k = \boldsymbol{z}_i + \alpha(\boldsymbol{z}_j - \boldsymbol{z}_i).$$

- *Identity* preservation $\Phi_\theta^{ii}(\boldsymbol{x}) = \boldsymbol{x}$:

$$\Phi_\theta^{ii}(\boldsymbol{x}) = \boldsymbol{x} + \int_0^1 \underbrace{\boldsymbol{f}_\theta^{ii}(\boldsymbol{x}, t)dt}_{=0} = \boldsymbol{x}.$$

**Implicit regularization: volume conservation.** By learning a divergence-free flow field for the deformation process, we show that the volume of any enclosed mesh can be conserved through the deformation sequence; see Proposition 4 (**supplementary material**). While we could penalize for divergence change via a loss, resulting in approximate volume conservation, we show how this hard-constraint can be implicitly and exactly satisfied without resorting to auxiliary loss functions. Based on Gauss's theorem, the volume integral of the flow divergence is equal to the surface integral of flux, which amounts to zero for solenoidal flows. Additional, any divergence-free vector field can be represented as the curl of a vector potential. This allows us to parameterize a *strictly* divergence-free flow field by first parameterizing a $C^2$ vector field as the vector potential. In particular, we parameterize the flow as $\boldsymbol{h}_\eta(\cdot) = \nabla \times \boldsymbol{g}_\eta(\cdot)$, with $\boldsymbol{g}_\eta$ using a fully-connected network: $\mathbb{R}^{3+c} \mapsto \mathbb{R}^3$. Since the curl operator $\nabla\times$ is a series of first-order spatial derivatives, it can be efficiently calculated via a sum of the first-order derivatives with respect to the input layer for $(x, y, z)$, computed through a single backpropagation step; refer to the architecture in Sec. B.1 (**supplementary material**).

**Implicit regularization: symmetries.** Given that many geometric objects have a natural plane/axis/point of symmetry, being able to enforce implicit symmetry is a desired quality for the deformation network. We can parameterize the flow function $\boldsymbol{h}_\eta(\cdot)$ by first parameterizing a $\boldsymbol{g}_\eta : \mathbb{R}^{d+1} \mapsto \mathbb{R}^d$. Without loss of generality, assume $d = 3$, and let $yz$ be the plane of symmetry:

$$\begin{cases} \boldsymbol{h}_\eta^{(x)}(x, y, z, t) = (\boldsymbol{g}_\eta^{(x)}(x, y, z, t) - \boldsymbol{g}_\eta^{(x)}(-x, y, z, t))/2 & \text{[anti-symmetric part]}, \\ \boldsymbol{h}_\eta^{(y,z)}(x, y, z, t) = (\boldsymbol{g}_\eta^{(y,z)}(x, y, z, t) + \boldsymbol{g}_\eta^{(y,z)}(-x, y, z, t))/2 & \text{[symmetric part]}, \end{cases} \quad (6)$$

where the superscript denotes the $x, y, z$ components of the vector output.

**Explicit regularization: surface metrics.** Additionally, surface metrics such as rigidity and isometry can be explicitly enforced via an auxiliary loss term to the overall loss function. A simple isometry constraint can be enforced by penalizing the change in edge lengths of the original mesh through the transformations, similar to the stretch regularization in [59, 60].

**Implementation.** We use a modified version of IM-NET [11] as the backbone flow model where we adjust the model with different number of hidden and output nodes. We defer discussions about the model architecture and training details to Sec. B.1 (**supplementary material**)

## 3.2  Hub-and-spoke deformation

Given a set of $N$ training shapes: $\mathcal{X} = \{X_1, X_2, \cdots, X_N\}$, we train the deformer by picking random pairs of shapes from the set. There are two strategies for learning the deformation, either by directly deforming between each pair of shapes, or deforming each pair of shapes via a canonical latent shape corresponding to a "hub" latent code. Additionally, we use an encoder-less approach (i.e., an auto-decoder [39]) where we initialize $N$ random latent codes $\mathcal{Z} = \{\boldsymbol{z}_1, \cdots, \boldsymbol{z}_N\}$ from $\mathcal{N}(0, 0.1)$,

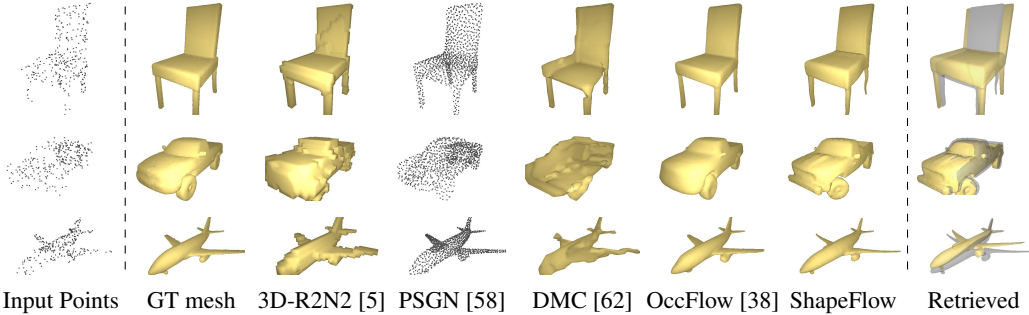

| Input Points | GT mesh | 3D-R2N2 [5] | PSGN [58] | DMC [62] | OccFlow [38] | ShapeFlow | Retrieved |

Figure 2: Qualitative comparison of mesh reconstruction from sparse point clouds as inputs. The shapes generated by ShapeFlow are able to preserve CAD-like geometric features (i.e. style) while faithfully aligning with the input observations (i.e. structure). The retrieved model is overlaid with the deformed model.

corresponding to each training shape. $\forall z \in \mathcal{Z}, z \in \mathbb{R}^c$. The latent codes are jointly optimized, along with the network parameters $\theta$. Additionally, we define a "hub" latent vector as $z_0 = [0, \cdots, 0] \in \mathbb{R}^d$. Under the hub-and-spokes deformation model, the training process amounts to finding:

$$\underset{\theta, \mathcal{Z}}{\arg\min} \sum_{(i,j) \in \mathcal{X} \times \mathcal{X}} \mathcal{C}(\Phi_\theta^{0j}(\Phi_\theta^{i0}(X_i)), X_j) + \mathcal{C}(X_i, \Phi_\theta^{0i}(\Phi_\theta^{j0}(X_j))). \tag{7}$$

A visualization for the learned latent space via the hub-and-spokes model is shown in Fig. 1(b). With hub-and-spokes training, we can define the sign function $s(\cdot)$ (Sec. 3.1) simply to produce $+1$ for the path towards the zero hub and $-1$ for the path from the hub, without the need of parameters.

### 3.3 Encoder-free embedding

We adopt an encoder-free scheme for learning the deformation space, as well as embedding new observations into the deformation space. After we acquire a learned deformation space by training with the hub-and-spokes approach, we are able to embed *new* observations of point clouds into the learned latent space by optimizing for the latent code that minimizes the deformation error of random shapes in the original deformation space to the new observation. Again, this "embedding via optimization" approach is similar to the auto-decoder approach in [39, 61]. The embedding $z_n$ of a new point cloud $X_n$ amounts to seeking:

$$\underset{z_n}{\arg\min} \sum_{i \in \mathcal{X}} \mathcal{C}(\Phi_\theta^{0i}(\Phi_\theta^{n0}(X_n)), X_i) + \mathcal{C}(X_n, \Phi_\theta^{0n}(\Phi_\theta^{i0}(X_i))). \tag{8}$$

## 4 Experiments

### 4.1 ShapeNet deformation space

As a first experiment, we learn the deformation space for entire classes of shapes from ShapeNet [4], and illustrate two downstream applications for such a deformation space: shape generation by deformation, and shape canonicalization. Specifically, we experiment on three representative shape categories in ShapeNet: chair, airplane and car. For each category, we follow the official train/test/validation split for the data. We preprocess the geometries into watertight manifolds using the preprocessing pipeline in [38], and further simplify the meshes to $1/10$th of the original number of vertices using [63]. The deformation space is learned by deforming random pairs of objects using a hub-and-spokes deformation approach (as described in Section 3.2). More training details for learning the deformation space can be found in Section B.2 (**supplementary material**).

#### 4.1.1 Surface reconstruction by template deformation

The learned deformation space can be used for reconstructing objects based on input observations. A schematic for this process is provided in Fig. 1: a new observation $X_n$, in the form of a point cloud, can be embedded into a latent code $z_n$ the latent deformation space according to Eqn. 8. The top-$k$

| category | Chamfer-$L_1$ ($\downarrow$) | | | | | IoU ($\uparrow$) | | | | | Normal Consistency ($\uparrow$) | | | | |
|---|---|---|---|---|---|---|---|---|---|---|---|---|---|---|---|
| | DMC | OccFlow | PSGN | R2N2 | ShapeFlow | DMC | OccFlow | PSGN | R2N2 | ShapeFlow | DMC | OccFlow | PSGN | R2N2 | ShapeFlow |
| airplane | 0.0969 | 0.0711 | 0.0976 | 0.1525 | 0.0858 | 0.5762 | 0.7158 | - | 0.4453 | 0.6156 | 0.8134 | 0.8857 | - | 0.6546 | 0.8387 |
| car | 0.1729 | 0.1218 | 0.1294 | 0.1949 | 0.1388 | 0.7182 | 0.8029 | - | 0.6728 | 0.6644 | 0.8222 | 0.8647 | - | 0.6979 | 0.7690 |
| chair | 0.1284 | 0.1302 | 0.1756 | 0.1851 | 0.1888 | 0.6250 | 0.6513 | - | 0.5166 | 0.4390 | 0.8348 | 0.8593 | - | 0.6599 | 0.7647 |
| mean | 0.1328 | 0.1077 | 0.1342 | 0.1775 | 0.1378 | 0.6398 | 0.7233 | - | 0.5449 | 0.5730 | 0.8235 | 0.8699 | - | 0.6708 | 0.7908 |

Table 1: Quantitative evaluation of shape reconstruction performance for ShapeNet [4] models.

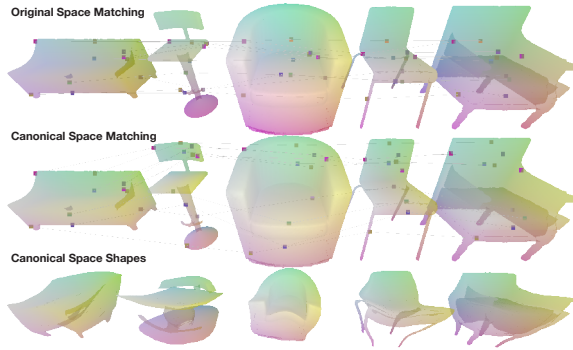
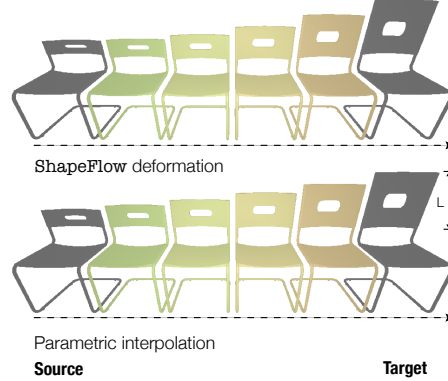

Figure 3: Unsupervised correspondences. Shapes RGB colors correspond to the $x, y, z$ coordinates of each point in the original / canonical space.

Figure 4: Deformation via parametric controllers [64] compared to the `ShapeFlow` interpolation.

nearest training shapes in the latent space are retrieved, and deformed to $z_n$. During this step we further fine tune the network parameters $\theta$ to perform a better fitting to the observed point cloud.

**Task definition.** We seek to reconstruct a complete object given a (potentially incomplete) sparse input point cloud. Following [38], we subsample 300 points from mesh surfaces and add a Gaussian noise of $0.05$ to the point samples. As a measure of the reconstruction quality, we measure the volumetric Intersection-over-Union (IoU), Chamfer-$L_1$, as well as normal consistency metrics.

**Results.** We benchmark against various state-of-the-art shape generation models that outputs voxel grids (3D-R2N2 [5]), upsampled point sets (PSGN [58]), mesh surfaces (DMC [62]) and implicit surfaces (OccFlow [38]); see quantitative results in Table 1. Qualitative comparisons between the generated geometries are illustrated in Figure 2. Note our shape deformations are more constrained (i.e., less expressive) than traditional auto-encoding/decoding, resulting in slightly lower metrics (Table 1). However, `ShapeFlow` is able to produce visually appealing results (Figure 2), as the retrieved shapes are of CAD quality – and *fine geometric details are preserved* by the deformation.

### 4.1.2 Canonicalization of shapes

An additional property of the deformation space learned through the hub-and-spoke formulation is that it naturally learns an aligned canonical deformation of all shapes. The canonical deformation corresponds to the zero latent code that corresponds to the hub, for shape $i : \{z_i, X_i\}$ it is simply the deformation of $X_i$ from latent code $z_i$ to the hub latent code $z_0 : \Phi_\theta^{i0}(X_i)$. Dense correspondences between shapes can be acquired by searching for the nearest point on the opposing shape in the canonical space. For a point $x \in X_i$, the corresponding point on $X_j$ is found as:

$$\psi^{i \to j}(x \in X_i) = \underset{y \in X_j}{\arg\min} ||\Phi_\theta^{j0}(x) - \Phi_\theta^{j0}(y)||_2^2. \tag{9}$$

**Evaluation metric.** To quantitatively evaluate the quality of the such surface correspondences learned in an unsupervised manner, we propose the *Semantic Matching Score* (SMS) as a metric for evaluating such correspondences. While semantic correspondences between shapes do not exist, semantic part labels are provided in various shape datasets, including ShapeNet. Denote $L(x)$ as an evaluation of the semantic label for the point $x$, $\langle \cdot, \cdot \rangle$ is a label comparison operator that evaluates to

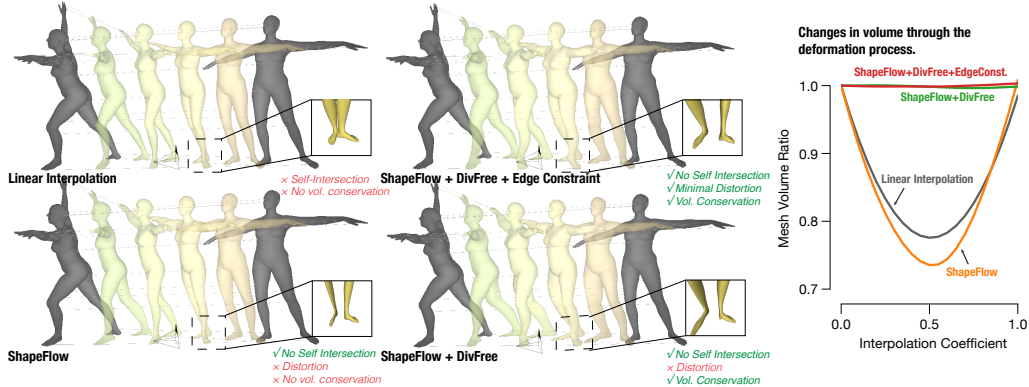

Figure 5: Animation of human figures via `ShapeFlow` deformation. (left) Intermediate poses interpolated between frames. (right) Volume change of the intermediate meshes, showing that our divergence-free flows conserve volume throughout the deformation.

one if the categorical labels are the same and zero otherwise. We define SMS between $(X_i, X_j)$ as:

$$\mathcal{S}(X_i, X_j) = \frac{1}{2}\Big( \frac{1}{|X_i|} \sum_{\boldsymbol{x} \in X_i} \langle L(\boldsymbol{x}), L(\psi^{i \to j}(\boldsymbol{x})) \rangle + \frac{1}{|X_j|} \sum_{\boldsymbol{x} \in X_j} \langle L(\boldsymbol{x}), L(\psi^{j \to i}(\boldsymbol{x})) \rangle \Big) \quad (10)$$

We choose 10,000 random pairs of shapes in the chair category to compute semantic matching scores.

**Results.** We first visualize the canonicalization and surface correspondences of shapes in the deformation space in Fig. 3. We compare the semantic matching score for our learned dense correspondence function with the naive baseline of nearest neighbor matching in the original

| Domain | SMS↑ |
|---|---|
| ShapeNet | 0.779 |
| ShapeFlow | **0.816** |

(ShapeNet) shape space. The results are presented in the inset table. The shapes align better in the canonical pose, and the matches found by canonical space matching are more semantically correct, especially between originally poorly aligned space due to different aspect ratios (e.g., couch and bar stool). This is reflected in the improved SMS matching score, as reported in the inset table.

## 4.2 Human deformation animation

`ShapeFlow` can be used to producing smooth animated deformations between pairs of 3D geometries. These animations are subject to the implicit and explicit constraints for volume and isometry conservation; see Section 3.1. To test the quality of such animated deformations, we choose two relatively distinct SMLP poses [2], and produce continuous deformations for in-between frames. Given that dense correspondences between shapes are given, we change the distance metric $\mathcal{C}$ in Eqn. 2 to be the pairwise $L_2$ norm between all vertices. We supervise the deformation with 5 intermediate frames produced via linear interpolation. Denoting the geometries at the two end-points as $i = 0$ and $j = 1$, the deformation at intermediate step $\alpha$ is:

$$X_{\alpha \in [0,1]} = \frac{1}{2}\Big( \Phi_\theta^{0\alpha}(X_0) + \Phi_\theta^{1\alpha}(X_1) \Big), \ z_\alpha = (1 - \alpha)z_0 + \alpha z_1 \quad (11)$$

**Results.** We present the results of this deformation in Figure 5. We compare several cases, including direct linear interpolation, deformation using an unconstrained flow, volume constrained flow, as well as volume and edge length constrained flow model. The volume change curve in Figure 5 empirically validates our theoretical result in Section 3.1, that (1) a divergence-free flow conserves the *volume* of a mesh through the deformation process, and (2) prevents self-intersections of the mesh, as in the example in Figure 5. Furthermore, we find that explicit constraints, such as the edge length constraint, reduces surface distortions.

## 4.3 Comparison with parametric deformations

As a final experiment, we compare the unsupervised deformation acquired using `ShapeFlow` with interpolations of parametric CAD models. We use an exemplar parametric CAD model from [64];

see Figure 4. `ShapeFlow` produces novel intermediate shapes of CAD level geometric quality that are consistent with those produced by interpolating a parametric model.

## 5  Conclusions and future work

`ShapeFlow` is a flow-based model capable to build high-quality shape-spaces by using deformation flows. We analytically show that `ShapeFlow` prevents self-intersections, and provide ways to regularize volume, isometry, and symmetry. `ShapeFlow` can be applied to reconstruct new shapes via the deformation of existing templates. A main limitation for the current framework is that it does not incorporate semantic supervision for matching shapes. Future directions include analyzing part structures of geometries by grouping similar vector fields [65], and exploring semantics-aware deformations. Furthermore, `ShapeFlow` may be used for the inverse problem of inferring a solenoidal flow field given tracer observations [66], an important problem in engineering physics.

## Acknowledgements

We thank Or Litany, Tolga Birdal, Yueqi Duan, Kaichun Mo for helpful discussions regarding the project. Andrea Tagliasacchi is funded by NSERC Discovery grant RGPIN-2016-05786, NSERC Collaborative Research and Development grant CRDPJ 537560-18, and NSERC Research Tool Instruments RTI-16-2018. The authors acknowledge the support of a Vannevar Bush Faculty fellowship, a grant form the Samsung GRO program, and gifts from Amazon AWS, Autodesk and Snap.

## Broader impact

The work has broad potential impact within the computer vision and graphics community, as it describes a novel methodology that enables a range of new applications, from animation to novel content creation. We have discussed the potential future directions the work could take in Sec. 5.

On the broader societal level, this work remains largely academic in nature, and does not pose foreseeable risks regarding defense, security, and other sensitive fields.

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
