[Supplementary Material]

# ShapeFlow
# Supplementary Material

## A   Mathematical proofs and derivations

**Proposition 1** (Intersection-free). *Any deformation map in $d$ spatial dimensions, $\Phi : \mathbb{R}^d \mapsto \mathbb{R}^d$, induced via a spatio-temporal continuous flow function $\boldsymbol{f} : \mathbb{R}^{d+1} \mapsto \mathbb{R}^d$, cannot induce self intersection of a continuous manifold $\Omega$ throughout the entire deformation process.*

*Proof.* Let $\boldsymbol{x}_a, \boldsymbol{x}_b \in \Omega$ be two points on the manifold, such that:

$$\boldsymbol{x}_a(t=0) \neq \boldsymbol{x}_b(t=0) \tag{12}$$

Assume that the two points intersect at $\boldsymbol{x}_a = \boldsymbol{x}_b = \tilde{\boldsymbol{x}}$ time $t = \tilde{t} \in (0,1]$. The location at time $t = 0$ can be found via:

$$\boldsymbol{x}_a(t=0) = \boldsymbol{x}_b(t=0) = \tilde{\boldsymbol{x}} + \int_{\tilde{t}}^{0} \boldsymbol{f}(\boldsymbol{x}(t), t) dt \tag{13}$$

which contradicts Eq. 12.   $\square$

**Proposition 2.** $\boldsymbol{x}_i(t) = \boldsymbol{x}_j(1-t)$ *is a sufficient condition for bijectivity.*

*Proof.* By respectively setting $t = 1$ and $t = 0$, we obtain:

$$\boldsymbol{x}_i(1) = \boldsymbol{x}_j(0) \Rightarrow \Phi_\theta^{ij}(\boldsymbol{x}_i) = \boldsymbol{x}_j \quad \text{for } t = 1 \tag{14}$$

$$\boldsymbol{x}_i(0) = \boldsymbol{x}_j(1) \Rightarrow \Phi_\theta^{ji}(\boldsymbol{x}_j) = \boldsymbol{x}_i \quad \text{for } t = 0 \tag{15}$$

We now replace $\boldsymbol{x}_j$ from (14) into (15) (and analogously for $\boldsymbol{x}_j$) showing that:

$$\Phi^{ij}(\Phi^{ji}(\boldsymbol{x}_j)) = \boldsymbol{x}_j \tag{16}$$

$$\Phi^{ji}(\Phi^{ij}(\boldsymbol{x}_i)) = \boldsymbol{x}_i \tag{17}$$

are nothing else than the bijectivity conditions.   $\square$

**Proposition 3** (Bijectivity condition on $\boldsymbol{f}$). *A sufficient condition on the flow function $\boldsymbol{f}_\theta$ for deformation bijectivity is $\boldsymbol{f}_\theta^{ji}(\boldsymbol{x}, t) = -\boldsymbol{f}_\theta^{ij}(\boldsymbol{x}, 1-t), \quad t \in (0, 1]$.*

*Proof.* We start by replacing (3) (and the equivalent for $ji$) into (16), and employing Proposition 2.

$$\left( \cancel{\boldsymbol{x}_i} + \int_0^1 \boldsymbol{f}_\theta^{ij}(\boldsymbol{x}_i(t), t) \, dt \right) + \int_0^1 \boldsymbol{f}_\theta^{ji}(\boldsymbol{x}_j(t), t) \, dt = \cancel{\boldsymbol{x}_i} \qquad (16) \leftarrow (3) \tag{18}$$

$$\int_0^1 \boldsymbol{f}_\theta^{ij}(\boldsymbol{x}_i(t), t) \, dt = \int_1^0 \boldsymbol{f}_\theta^{ji}(\boldsymbol{x}_j(t), t) \, dt \qquad [t \leftarrow (1-t)] \tag{19}$$

$$\int_0^1 \boldsymbol{f}_\theta^{ij}(\boldsymbol{x}_i(t), t) \, dt = -\int_0^1 \boldsymbol{f}_\theta^{ji}(\boldsymbol{x}_j(1-t), 1-t) \, dt \qquad [\boldsymbol{x}_i(t) = \boldsymbol{x}_j(1-t)] \tag{20}$$

$$\int_0^1 \boldsymbol{f}_\theta^{ij}(\boldsymbol{x}_i(t), t) \, dt = \int_0^1 -\boldsymbol{f}_\theta^{ji}(\boldsymbol{x}_i(t), 1-t) \, dt \tag{21}$$

Which is satisfied $\forall t \in (0, 1]$ and via the constraint:

$$\boldsymbol{f}_\theta^{ji}(\boldsymbol{x}, t) = -\boldsymbol{f}_\theta^{ij}(\boldsymbol{x}, 1 - t), \quad t \in (0, 1] \tag{22}$$

$\square$

**Proposition 4** (Volume conservation). *Suppose $V$ is a compact subset of $\mathbb{R}^d$. For $d = 3$, $V$ is the three-dimensional volume, and $\partial V$ is the surface boundary of $V$. Given a deformation map in $d$ spatial dimensions, $\Phi : \mathbb{R}^d \mapsto \mathbb{R}^d$, induced via a divergence-free (i.e. solenoidal) spatio-temporal continuous flow function $\boldsymbol{f} : \mathbb{R}^{d+1} \mapsto \mathbb{R}^d$, $\nabla \cdot \boldsymbol{f} = 0$, the volume within the deformed boundary $\Phi(\partial V)$ remains constant.*

*Proof.* As per divergence theorem, the flux $(\boldsymbol{f} \cdot \boldsymbol{n})$ across the boundary $\partial V$ integrates to zero:

$$\oiint_{\partial V} (\boldsymbol{f} \cdot \boldsymbol{n}) \, d\partial V = \iiint_V (\nabla \cdot \boldsymbol{f}) \, dV = 0 \tag{23}$$

$\square$

**Theorem 1** (Existence of vector potential). *If $\boldsymbol{f}$ is a $C^1$ vector field on $\mathbb{R}^3$ with $\nabla \cdot \boldsymbol{f} = 0$, then there exists a $C^2$ vector field $\boldsymbol{g}$ with $\boldsymbol{f} = \nabla \times \boldsymbol{g}$.*

*Proof.* This extends from the fundamental theorem of vector calculus, and is the result of the vector identity $\nabla \cdot (\nabla \times \boldsymbol{g}) = 0$. $\square$

# B  Implementation details

## B.1  Neural architecture

We employ a variant of the IM-NET [11] architecture as our backbone. The complexity of the flow model is parameterized by the number of feature layers $n_f$, as well as the dimensionality of the latent space $c$. In the case with no implicit regularization, we directly use the IM-NET backbone as the flow function $\boldsymbol{h}_\eta(\cdot)$ (in Eqn.6). In the case with implicit volume or symmetry regularization, we use the backbone to parameterize $\boldsymbol{g}_\eta(\cdot)$; see Figure 6 for a a schematic of the backbone. We do not learn an encoder, and instead we use an encoder-less scheme (Sec. 3.3) for training as well as embedding new observations into the deformation space.

Figure 6: Backbone flow architecture

## B.2  Training details

**ShapeNet deformation space (Sec. 4.1).** For training `ShapeFlow` to learn the ShapeNet deformation space, we use a backbone flow model with $n_f = 128$ feature layers, use the `ReLU` activation function, learning rate of $1e-3$, a batch size of $256$ (across 8 GPUs), and train for $204800$ steps. We train using an Adam Optimizer, and we compute the time integration using the dopri5 solver with relative and absolute error tolorence of $10^{-4}$. We samples $512$ points on each mesh to use as proxy for computing point-to-point distance. We enforce symmetry condition on the deformations. We do not enforce isometry and volume conservation conditions since they do not apply to the shape categories in ShapeNet. For the reconstruction experiment (Sec.4.1.1) we use latent dimensions of $c = 128$, as a compact latent space allows better clustering of similar geometries, improving retrieval quality. For the canonicalization experiment (Sec. 4.1.2), we use a $c = 512$, since a larger latent dimension mitigates distortions at the canonical pose.

Furthermore, after training the deformation space, for embedding a new latent code, we initialize the latent code from $\mathcal{N}(0, 10^{-4})$. We optimize using Adam optimizer with learning rate of $1e-2$ for 30 iterations. Then we finetune the neural network for the top-5 retrieved nearest neighbors, for an additional 30 iterations.

Figure 7: Random $5 \times 5$ examples of shapes in the deformation space. The diagonal shapes (in green) are the identity transformations for the shapes. The identity transformations are able to preserve the original geometric details almost perfectly, highlighting the identity preservation capability of `ShapeFlow`.

**Human deformation animation (Sec. 4.2).** For human model deformation, since we are only learning the flow function for two, or a couple of shapes, we can afford to use a more lightweight model. We use a backbone flow model with $n_f = 16$. We use the `Elu` activation, since it is $C^2$ continuous, allowing us to parameterize a volume-conserving divergence-free flow function. We use the Adam optimizer with a learning rate of $2e - 3$. For improved speed, we use the Runge-Kutta-4 (RK4) ODE solver, with 5 intermediate time steps. For the best performing result, we use the divergence-free parameterization, as well a edge loss weighting factor of $2.0$. We optimize for 1000 steps.

## C  Additional analysis and visualization

### C.1  Deformation examples

We show additional examples of the deformation between random pairs of shapes in the deformation space. We present the visualizations in Fig. 7. We draw random subsets of 5 shapes at a time, and plot the pairwise deformations of the shapes in a $5 \times 5$ grid. One takeaway from this is that when the source and target are identical, the transformation amounts to an identity transformation. By transforming the shape to and back from the "hub", the geometric details are almost exactly preserved.

### C.2  Effects of integration scheme

We further study the impacts of the ODE solver scheme on the shape deformation. We note that for the ShapeNet deformation space, it involves much more shapes ($N = 4746$) than the case of human frame interpolation, therefore it involves much drastic deformations. A fixed-step solver, such as the RK4 solver, is not able to accurate compute the dynamics for the individual points.

Figure 8: Deformation of shape examples in the deformation space via an RK4 ODE solver.

Numerical error accumulated during the integration step leads to violations of *non self-intersection*, *identity preservation*, resulting in dramatically unsatisfactory deformations between shapes.