[Reviews · NeurIPS 2020]

Review 1

Summary and Contributions: This paper proposes a mathematically sound learned shape flow method to deform shapes from (potentially sparse) observations to template shapes.

Strengths: - shape representation are an important topic in machine learning, computer vision and computer graphic - the method seems mathematically sound and allows to guarantee important properties during deformation

Weaknesses: - it is not clear how the initial alignment between shapes would in general be obtained so that this method can be used on scanned data in the wild.

Correctness: - They seem to be, but didn't check in detail.

Clarity: yes.

Relation to Prior Work: yes.

Reproducibility: Yes

Additional Feedback:


Review 2

Summary and Contributions: The authors propose ShapeFlow, a flow-based model for learning a deformation space for 3D shapes.The authors recognize the weaknesses of either using a template-based method or geometry encoder-decoder, and propose the different approach of representing a geometry in a data-driven fashion. The authors prove their proposed method has desired properties critical to the quality of representation, and give plausible results. The authors show their method can be used in many applications.

Strengths: 1. Novel contribution in defining deformation flow towards a data driven approach. The definition in formula (5) is simple and elegant, and shows great capabilities of representing the desired deformation. The authors take great considerations of potential applications when designing their method, and show their formulation is easy and natural to obtain certain properties towards specific needs. The theoretical proofing of the method is especially strong and appreciated. 2. Easy to implement. The network design for their proposed method is simple and straightforward. The idea of encoding a geometry using proposed flow could be beneficial in many other 3D related work. 3. Various applications. The authors show that proposed method can be applied to 3d reconstruction, template fitting, correspondence matching, template editing, and mesh interpolation. They provide plausible results and comparisons towards other methods. The idea of this work can also be important in follow-up studies of the canonical, the style and the structure of 3D data.

Weaknesses: 1. The proposed network takes the xyz coordinates of the mesh as part of the input feature. The authors did not specify what coordinate system do these xyz and flow definition reside in, though the reviewer expected they are taken in the canonical space of the meshes. Rotations, translations and scale of the meshes would greatly affect the values of such coordinates. It would be helpful if the authors could provide their insights in normalizing 3D space here, and if the method could distinguish deformation and global transformation of the meshes. 2. See section Correctness 2.

Correctness: The claims and method are mostly correct in this paper. The reviewer found these places where could need authors' attention: 1. line 50. The reviewer believe volumetric methods are computationally intensive because of the voxelization and evaluation on many 3D grid vertices. The statement sounds to the reviewer like marching cubes itself is so(marching cubes is fast). 2. line 159 under volume conservation. The proposed flow parameterization is proven volume conservative and strengthens the novelty of the paper. However, the reviewer would like to point out the set of parametric flow is a proper subset of all volume conservative flow. The statement is not strong on claiming hard-constraint because you only implicitly regulate towards a subset, while using a loss on divergence change would explicitly regulate all cases for volume conservation. After reading the rebuttal the reviewer still believes this part remains a weak statement.

Clarity: Yes. The reviewer found these places where could need authors' attention: 1. line 35. '...to shape generation based continuous flow...' Is it 'based on'? 2. formula (3) under line 133. The authors used lower case x_i for both the enumerating element of set X_i and the 'distribution' function x_i(T). This was very confusing the first time read, and a change of either symbol is recommended by the reviewer.

Relation to Prior Work: Yes.

Reproducibility: Yes

Additional Feedback: This paper falls in the area of 'Applications -> Computer Vision' and 'Applications -> Body Pose, Face, and Gesture Analysis'. Due to its emphasis on the formulation the 3D flow and analysis on geometry processing, the reviewer believes it would also fit nicely under the area of Computer Graphics, and attracts great interests from the graphics community.


Review 3

Summary and Contributions: It is suggested to learn a NN and optimize a set of latent codes on a mesh dataset, such that the NN, when provided two of these codes, produces a deformation field to align the two meshes with these codes nicely. "Nicely" means, preserving volume, free of intersection, etc. This all is achieved by not learning the deformation directly, but a vector potential, from which the (differentiable) curl operator produces a deformation. These representations are linked using a time integration step. Provided a new shape, the method optimizes to find a new latent code to embed the new shape and, using the deformation network, can then be used to align nearby shapes or to deform into a canonical reference.

Strengths: -- It was enjoyable to read, with a few typos but a very nice non-verbose precise style and adequate use of math. -- The deformation indeed produces meshes that show some details. -- The idea to optimize derivatives and integrate these is attractive and I have not seen it before in ML. It is in agreement with what Graphics is doing since a long time, but not as novel from that perspective.

Weaknesses: -- It misses to cite they key papers from Graphics looking into such tasks -- The non-parametric aspect is oversold in the usual way as in any non-parametric paper: yes, the car has details. That is great. No, they are not "reconstructed". -- To embed a new shape, an optimization is required. What is the benefit of not having an encoder? -- Comparison to very trivial alternatives with similar non-parametric benefits are missing. For example vanilla nearest neighbour, either in Chamfer distance or in some embedded space learned in a conventional way, say, an auto-encoder. Nearest-neighboring is a trivial property of a good space. The paper should be proud of construction an embedding using deformation. Not of the ability to nearest-neighbour in it. -- To apply a deformation, costly integration has to be performed

Correctness: Yes In L220 an explanation is missing how the k meshes deformed are merged into one? Like this we will have a Baroque chair on top of a Bauhaus one that were both deformed to look similar to the input. That can't be no solution to anything.

Clarity: Yes. I very much enjoyed reading the technical part. On style: The only downside was the made-up praise of non-parametric aspects. It is not that the method "preserves" any details. I think authors know better and poor students and industry readers will suffer not benefit from such marketing. What happens is that the pick up truck has a few little details and fenders and screws which make it look quite cool. Alas this is not preserving anything. It just makes it up. Which is okay. But there is an important difference between that and reconstructing. Its the same as with GAN: a blobby car is not from the data distribution, but the random sample from the GAN is also not "preserving" anything. Similarly on style: The use of the word "disentanglement" (L7, already in the abstract, later L68 in the list of contributions) is unfortunate at best. What is disentangled here? When you take a noisy scan from a chair and you run the show, what you get back is a chair with some nice details. Sometimes Bauhaus, sometimes Baroque. That is not "disentangled". It is more like Forest Gump's box. You might get something, but not sure what it will be. If it was "disentangled" it could be controlled independently. So I can say I want the Bauhaus version. So maybe replace "disentagled" with "ignored" or "randomized"? L 34 is another such instance: "autoencoder removes the details". Aha. Ok. Fair enough. Are the details we get from the proposed method any better than getting the autoencoder result and doing a L2 or latent-space nearest-neighbor query in the training data and show those details instead? They should be called what they are: decoration. In L232 details are "preserved". Nothing is preserved. It is made-up. That is okay. But please say it. L37 I would not agree on the "novelty of the perspective". It is valid, but the two key ideas I was identifying (non-parametric and flow integration) are not novel to a learned reader. It was not understood why L120 ff introduces meshes and edges and all that. In the end they are not used. It would work equally well on point clouds, CT scans, maybe even images. As long as we have a deformation operator and a metric on them. The paper is very compact, making links to supplemental nicely. It could be said more general. L126 same thing: yes you retain connectivity. As it does not even matter when space is deformed what is inside. Could authors explain a use case of the invertibility in L140? I agree it feels nice from the math side, but is it actually used in the paper or usable in any conceivable application? When I want to go back, I keep the original mesh on my drive / in memory? I thought Eq 5 and the following derivations to be very nice. Daunting at first, but the exposition gave a very good guidance. Still I did not manage to understand the consistency and identity results. Is this desiderata or is this properties following from construction? At present it reads like proof-by-assumption: we want it to be identity and it is because it is. This might be related to L140 not being clear to me. Has the sketched idea (enforcing vector field props via the loss) been tried? What is the benefit? After all we save the integration compute, also at test time. What else can be said about the integration, how is it performed even? Newton? How many steps? Maybe just one, scaled by the latent distance? What is the error, etc. What would happen when unrolling the Newton into the learning? Do any of the results use L172? If yes, does this imply the learning data is aligned? I thought this to be the case in parts for ShapeNet. Fig. 1b did not show me what it sys it shows in L193. No hub, no spoke. Fig. 1 is not great. It is not up to the precision of the technical writing. L200 explains how you need to test-time optimize. I never understood why this is done, also in other papers. Was it attempted to have an encoder instead? What is the drawback? At this point, I note to not understand how large c is, the latent code dimension in the results. Maybe I missed it. In L211 it was not clear why meshes are subsampled. It would be easier to MC the Chamfer in the loss. Deforming a full-res mesh should not cost so much. Fig. 3 I found hard to parse visually. Less could be more. And larger handle points. L251 something is wrong with TeX I guess. Sec 4.3. did not have enough details for me to understand what has been done. How large is the dataset, how diverse, how many parameters. I can guess, but that is too vague.

Relation to Prior Work: This is a substantial problem. The entire reasoning about how working in a derived domain that can later be integrated is just von Funck 2006 implemented in TensorFlow. @article{vonFunck2006vector, title={Vector field based shape deformations}, author={Von Funck, Wolfram and Theisel, Holger and Seidel, Hans-Peter}, journal={ACM Transactions on Graphics (TOG)}, volume={25}, number={3}, pages={1118--1125}, year={2006} } Sure there is more, like latent codes and all this. Also I did not find again (sorry), but felt pretty confident, that there is plenty of papers that take image or shape collections and align them using deformations. The paper instead cites some popular AI work that is maybe deep, but not very founded or even functional in terms of the resulting deformations. Sec. 4.2. von Funck and many follow up works will do righ away. Also the story that morphable model -> same topology is plain wrong. That is what Blanz and Vetter did to make it 3D in 90ies conditions. But the default way to do this now is decoupled from the domain. So Blanz and Vetter used the same mesh they deform to represent the space. Later, everybody used much finer meshes with entirely different topology.

Reproducibility: Yes

Additional Feedback:

[Author Response · NeurIPS 2020]

First, we thank the reviewers for their detailed, constructive, and positive feedback on the paper. We are happy to see the they appreciate our contributions in proposing a method that is "mathematically sound" [R1], "simple and elegant" [R3], and "enjoyable to read" [R4], while addressing "an important topic in machine learning, computer vision and computer graphics" [R1]. Below we address specific feedback from the reviewers.

**Geometry alignment in the normalized frame [R1, R3].** In the ShapeNet deformation space experiment, learning happens in the normalized shape frame (as provided by ShapeNet), which is similar to most works in the literature concerning 3D geometry generation. The inherent alignment in this normalized space can facilitate a more meaningful geometric loss via the symmetric Chamfer distance when considering deformation between objects. However, as we illustrate in experiment 4.1.2: Canonicalization of Shapes, by deforming through a common "hub" shape, the deformation can further regularize the shapes in this space to achieve much better alignment in the deformed canonical shape space – thus providing more semantically meaningful correspondences between geometries. For extending this method in real-world settings, one can envision a two-step approach that combines an initial pose estimation algorithm with the "reconstruction via deformation" approach proposed in this work.

**Disentangling deformation and global transformation [R3].** The deformation methodology proposed in this work uses a single flow field to serve as the deformation function, which includes all transformations, global or local. It works well when dense correspondences are provided (e.g., Figure 5 for the animation experiment). To explicitly decouple global and local transforms, one can learn composed flow fields of pure rotational flow (parameterized with 3 DOF), pure translational flow (parameterized with 3 DOF), and local deformation flow (parameterized with a neural network). We leave more detailed investigation of this approach to future studies.

**Related work in Graphics [R4].** We thank the reviewer for pointing out the work of Von Funck et al., we will cite that in the camera-ready version of this work.

**Accuracy of statements [R3, R4].** (1) Regarding the discussion about morphable models, we note in our paper that morphable models are commonly used for shapes with small intra-class variations, and generally assume a shared topology (at inference time, not training), which we believe is an accurate statement. We would kindly ask the reviewer to provide a reference to a paper where this statement does not hold. (2) Regarding the claim about detail preservation, there is only one instance where we mentioned that, in L232, in the context that "details are preserved by the deformation." We are not claiming that detail is preserved from the input point cloud to the output geometry. (3) *Marching Cubes is fast* – Yes, other aspects of volumetric methods are less efficient (sampling for OccNet/DeepSDF, high memory usage for voxel methods, etc.). We will rephrase L50 to be more accurate. (4) *You only implicitly regulate towards a subset, while using a loss on divergence change would explicitly regulate all cases for volume conservation.* – We agree with the first statement, but even using a loss on divergence, the flow itself is still just a parametric flow.

**Miscellaneous questions [R4].** (1) *Why not use an encoder?* – an encoderless scheme is more flexible and forgiving regarding partial and incomplete inputs, whereas a scheme that uses an encoder will require a substantial amount of data augmentation to train – it is just a design choice. (2) *Comparison to direct nearest neighbor retrieval* – an example of the directly retrieved geometry is given in Figure 2, rightmost column. (3) *To apply a deformation, costly integration has to be performed* – the integration step executed on a GPU is fast, «1sec. (4) *How are the $k$ meshes deformed merged into one* – we pick the best one; we will add a more detailed description in L210. (5) *Why introduce meshes and edges in L120* – mesh structure of the source is constant in the deformation process, with the manifold property preserved in the process. (6) *Why do you need invertibility* – invertibility of the space facilitates training. For instance, training $A \rightarrow B$ automatically trains $B \rightarrow A$. Moreover, training $A \rightarrow H$ (hub) facilitates the training of $* \rightarrow H \rightarrow A$. (7) *Has the sketched idea (enforcing vector field props via the loss) been tried?* – No, but this will require computing loss on random volumetric samples, which quickly becomes expensive, and does not guarantee constraint satisfaction. (8) *Do any of the results use L172?* – Yes, in the ShapeNet experiment. (9) *In L211 it was not clear why meshes are subsampled* – for training a deformation space, we need to batch multiple samples together, and subsampling makes training more efficient; we do not subsample at test time.

**Typos, math notation issues, LaTeX glitch [R3, R4].** Thanks, we will address them in the camera ready version!

[Meta-Review · NeurIPS 2020]

The three expert reviewers agree on the merits of the submission, as it introduces and validates an interesting and elegant approach that addresses an important and challenging problem of learning deformable models from data. Based on the feedback from the reviewers, the acceptance is recommended, potentially as a spotlight. A side note: the authors rightfully draw connection of their approach to the auto-decoder model [39]. The authors of [39], however, missed an earlier work of Bojanowski et al. ICML18 (Optimizing the Latent Space of Generative Networks) that had previously introduced and popularized "encoder-free models". This lead to unfortunate fragmentation/duplication of terminology. The authors are encouraged to consider the connection to and the terminology from Bojanowski et al. when discussing "auto-decoders"/"encoder-free models" in the final version.